# Anterior transversalis fascia approach versus preperitoneal space approach for inguinal hernia repair in residents in northern China: study protocol for a prospective, multicentre, randomised, controlled trial

Qing Fan,[1] De-wei Zhang,[1] Da-ye Yang,[1] Hong-wu Li,[1] Shi-bo Wei,[1] Liang Yang,[1] Fu-quan Yang,[2] Shao-jun Zhang,[3] Yao-qiang Wu,[4] Wei-de An,[5] Zhong-shu Dai,[6] Hui-yong Jiang,[7] Fu-rong Wang,[8] Shi-feng Qiao,[9] Hang-yu Li[1]

For numbered affiliations see end of article.

**Correspondence to**
Dr Hang-yu Li;
li_hangyu@126.com

## ABSTRACT

**Introduction** Many surgical techniques have been used to repair abdominal wall defects in the inguinal region based on the anatomic characteristics of this region and can be categorised as 'tension' repair or 'tension-free' repair. Tension-free repair is the preferred technique for inguinal hernia repair. Tension-free repair of inguinal hernia can be performed through either the anterior transversalis fascia approach or the preperitoneal space approach. There are few large sample, randomised controlled trials investigating the curative effects of the anterior transversalis fascia approach versus the preperitoneal space approach for inguinal hernia repair in patients in northern China.

**Methods and analysis** This will be a prospective, large sample, multicentre, randomised, controlled trial. Registration date is 1 December 2016. Actual study start date is 6 February 2017. Estimated study completion date is June 2020. A cohort of over 720 patients with inguinal hernias will be recruited from nine institutions in Liaoning Province, China. Patient randomisation will be stratified by centre to undergo inguinal hernia repair via the anterior transversalis fascia approach or the preperitoneal approach. Primary and secondary outcome assessments will be performed at baseline (prior to surgery), predischarge and at postoperative 1 week, 1 month, 3 months, 1 year and 2 years. The primary outcome is the incidence of postoperative chronic inguinal pain. The secondary outcome is postoperative complications (including rates of wound infection, haematoma, seroma and hernia recurrence).

**Ethics and dissemination** This trial will be conducted in accordance with the Declaration of Helsinki and supervised by the institutional review board of the Fourth Affiliated Hospital of China Medical University (approval number 2015–027). All patients will receive information about the trial in verbal and written forms and will give informed consent before enrolment. The results will be published in peer-reviewed journals or disseminated through conference presentations.

## Strengths and limitations of this study

► This trial will be the first prospective multicentre randomised controlled study involving nine institutions in Liaoning province of Northern China to provide reliable results from a representative study population.
► This trial will compare postoperative complications after anterior transversalis fascia approach versus preperitoneal space approach for inguinal hernia repair in residents in Northern China.
► Based on ethical and economic considerations, interim analysis will be performed during the trial to reduce unnecessary waste of manpower and materials (lower cost).
► The limited time for studying postoperative recurrence of hernia likely influences the judgement of long-term recurrence of inguinal hernia.
► Variability of surgeons from each study centre could be a major confounder to bias the results of this trial.

**Trial registration number** NCT02984917; preresults.

## BACKGROUND
### History and current related studies

Inguinal hernia is a common surgical disease that manifests as protrusion of abdominal cavity contents through the inguinal canal because of an abdominal wall defect. It is more common in men than in women, with an overall incidence of 5%–10%.[1] Methods for surgical repair of abdominal wall defects in the inguinal region are classified as either 'tension' repairs or 'tension-free' repairs. Herniorrhaphy through repair of the posterior wall of the inguinal canal was first

described by Bassini in 1887 and is regarded as a classic surgical method.[2]

As understanding of the anatomic location and patho-physical characteristics of inguinal hernia developed, the American surgeon Lichtenstein proposed a new concept of tension-free herniorrhaphy.[3] This technique was quickly adopted worldwide because of its advantages including minimal invasion, technical ease, effectiveness, low complication rate, low recurrence rate and allowance of resumption of unrestricted physical activity. The most common technique is open tension-free herniorrhaphy.

Tension-free herniorrhaphy methods include anterior transversalis fascia repair, preperitoneal repair, abdominal cavity patch repair and combined repair approaches.[4–6] Lichtenstein herniorrhaphy is the representative technique of anterior transversalis fascia repair. Preperitoneal repair techniques include transabdominal preperitoneal, total extraperitoneal and Kugel repair techniques. The combined repair approaches refer to tension-free herniorrhaphy using a modified Kugel patch and the Ultrapro hernia system.[7 8]

Many surgical repair methods involving patches (of varying types and materials) are available for inguinal hernia repair. Zhu *et al*[9] performed a meta-analysis regarding open extraperitoneal approach and extraperitoneal laparoscopic hernioplasty for inguinal hernia repair. They found that these two approaches exhibited basically similar clinical outcomes. Patients receiving extraperitoneal laparoscopic hernioplasty needed shorter hospital stays and exhibited lower incidence of postoperative complications. Patients receiving open extraperitoneal approach exhibited lower incidence of peritoneal tears. Pisanu *et al*[10] analysed the clinical efficacy of laparoscopic and Lichtenstein techniques in recurrent inguinal hernia repair. They found that laparoscopicy showed lower incidence of chronic inguinal pain and an earlier return to normal daily activities but greatly longer operative time. There are many randomised controlled trials[11–15] on the clinical efficacy of inguinal hernia repair approaches (table 1), but little is reported on anterior transversalis fascia approach and preperitoneal space approach for inguinal hernia repair in residents in Northern China.

There are few reports on the effects of different inguinal hernia repair approaches on postoperative complications, particularly regarding severe postoperative complications, in patients in northern China. The incidence of severe complications after inguinal hernia repair is relatively low, but this surgery can lead to physical impairment or organ dysfunction that greatly decreases patient quality of life and places a heavy burden on the patient's family and society. At present, there is a scarcity of clinical evidence from large sample, randomised controlled trials investigating the effectiveness of inguinal hernia repair via the anterior transversalis fascia approach versus the preperitoneal approach.

## Main objectives
In this study, we will investigate the advantages and disadvantages of the anterior transversalis fascia approach versus the preperitoneal approach for inguinal hernia repair in residents from northern China regarding common postoperative complications (including acute and chronic pain, wound infection, rates of wound infection, haematoma, seroma and hernia recurrence) and severe postoperative complications. These outcomes will provide trial-based evidence for selection of rational therapeutic regimen in the clinic.

## Distinguishing features from related studies
(1) This study will use centre-based stratification to investigate the effects of anterior transversalis fascia approach versus preperitoneal space approach for inguinal hernia repair on postoperative chronic inguigal pain and other common complications. This study will determine the optimal surgical hernia repair approach that is suitable for the anatomic characteristics of the inguinal region of residents in northern China and corresponds to the regional economic conditions. (2) Cost-utility analysis will be analysed using centre-based stratification. (3) To analyse the effects of different surgical repair approaches (involving various patch types and materials) on postoperative quality of life.

## Methods/Design
### Study design
This is a prospective, large-sample, multicentre, randomised controlled trial that will include a cohort of over 720 patients with inguinal hernia. The trial committee organisation and contributions and role are provided in the see online supplementary file 1. In strict accordance with the Consolidated Standards of Reporting Trials standards,[16] the baseline data (preoperative data), therapeutic regimen, therapeutic outcome and medical costs during hospitalisation of patients with inguinal hernia will be recorded. Patient data will be collected using an electronic data capture system (EDC).

According to recommendations for treatment and follow-up of inguinal hernia repair in adults in the Guidelines for the Diagnosis and Treatment of Inguinal Hernia in Adults (2014 Edition) formulated by Chinese scholars[17] and the Adult Inguinal Hernia Treatment Guidelines (Updated Edition in 2014) developed by the European Hernia Society,[18] patients who undergo herniorrhaphy will be followed up at seven time-points: at baseline (at admission, visit 1), predischarge and at 1 week (visit 2), 1 month (visit 3), 3 months (visit 4, clinic visit or telephone follow-up), 1 year (visit 5, telephone follow-up) and 2 years after surgery (visit 6, telephone follow-up). The flow chart of the study protocol is shown in figure 1. Prior to surgery: patients will be rescreened against inclusion and exclusion criteria. Signed informed consent will be obtained. Patient's demographic data, history of disease and medication and admission condition and vital signs will be recorded. Clinical examination data

**Table 1** RCTs regarding inguinal hernia repair approaches

| Study | Design | Subjects | Disease | Follow-up time | Outcome measures | Conclusion |
|-------|--------|----------|---------|----------------|------------------|------------|
| Akhtar et al.[11] | RCT | TAPP (n=30) Lichtenstein (n=50) | Unilateral inguinal hernia | 6 months | Average operation, pain score, analgesics, admission days, days required to return to work | Laparoscopic hernia surgery is better than Lichtenstein repair in terms of postoperative pain, hospital stay and return to daily activity. |
| Sarhan et al.[12] | RCT | A total of 200 patients scheduled for unilateral inguinal hernia repair were randomly divided into two groups to undergo either laparoscopic TAPP (group A) or open modified Kugel procedure | Unilateral inguinal hernia | 32 months | Recurrence and short-term and long-term complications | Both open modified Kugel and laparoscopic TAPP preperitoneal repair techniques for inguinal hernia are safe and effective with low recurrence rates. Laparoscopic approach has better outcome in terms of chronic pain, short operative time and short duration of hospital stays. |
| Kargar et al.[13] | RCT | TAPP (n=60) Lichtenstein (n=60) | Inguinal herniajrnlTblFoot | Follow-up occurred within 6 weeks. | Pain score (VAS), haematoma/seroma, urinary retention, wound infection, hospital stay | The laparoscopic TAPP repair is safer and less complicated approach for inguinal hernia repair. The two main short-term advantages of the laparoscopic TAPP repair with the tension free Lichtenstein repair were less postoperative pain and earlier return to the normal life activities. No difference was seen in overall complications. |
| Salma et al.[14] | RCT | TAPP (n=30) Lichtenstein (n=30) | Direct inguinal hernia | Postoperative pain intensity assessed by VAS and hospital stay measured in hours. | Hospital stay, immediate post operative pain | There is less postoperative pain after laparoscopic repair but hospital stay is same in both the procedures but laparoscopic procedure does increase the cost. |
| Bahram[15] | RCT | TAPP (n=150) Lichtenstein (n=150) | Inguinal herniajrnlTblFoot | Three hundred patients with inguinal hernia were enrolled in this study, divided into two equal groups: Group I managed by TAPP laparoscopic repair and group II managed by open lichtenstein repair. | Operative time, intraoperative visceral injury, ileus, hospital stay or wound complications, postoperative pain, groin hypothaesia, return to activities, recurrence | TAPP technique is an excellent approach for treatment of inguinal hernia in comparison to LR either unilateral or bilateral, primary or recurrent inguinal hernia with low morbidity and recurrence comparable to that oflichtenstein repair with advantages of less postoperative pain and early return to activities. |

LR, laparoscopic repair; RCT, randomised controlled trial; TAPP, transabdominal preperitoneal; VAS, Visual Analogue Scale.

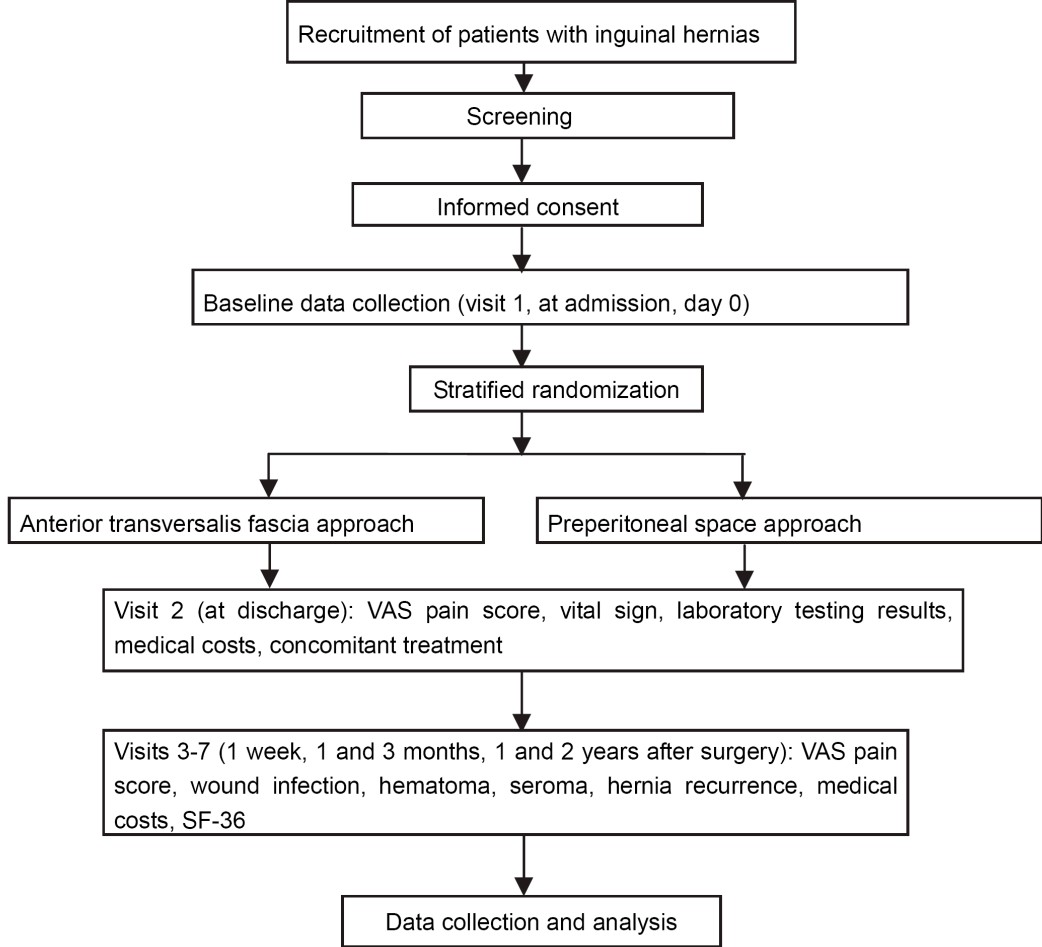

**Figure 1** Flow chart of study protocol. VAS, Visual Analogue Scale; SF-36, 36-Item Short Form Health Survey.

will be collected from each centre, including history of disease, physical examination, laboratory testing results, imaging findings, preoperative VAS pain score, intraoperative findings and details of occurrence and management methods of intraoperative injury to the intestinal tract and bladder, spermatic cord and vascular system.

► Predischarge, and 1 week, 1 and 3 months, 1 and 2 years after surgery: pain, wound infection, haematoma and seroma in the inguinal region and hernia recurrence will be recorded. Medical costs during hospitalisation and patient quality of life after discharge will be also recorded. The flow chart of study protocol is shown in figure 1. And The Standard Protocol Items: Recommendations for Interventional trials (SPIRIT checklist (Standard Protocol Items: Recommendations for Interventional Trials)) was followed in designing the study protocol (see online supplementary file 2).

## Patients

Patients with inguinal hernia will be recruited from nine trial centres in northern China: the Department of General Surgery, the Fourth Affiliated Hospital of China Medical University; Department of General Surgery, Branch 3, First Hospital of Dalian Medical University; Department of General Surgery, the 202 Hospital of Chinese PLA; Ward of Hernia, Department of General and Gastrointestinal Surgery, First Affiliated Hospital of Liaoning Medical University; Department of General Surgery, General Hospital of Shenyang Military; Second Department of General Surgery, General Hospital of Benxi Iron and Steel Co, Ltd; First Department of General Surgery, Affiliated Central Hospital of Shenyang Medical University; Department of General Surgery, First Hospital of Dandong; Shengjing Hospital of China Medical University.

## Inclusion criteria

Male patients presenting with all of the following conditions will be considered for study admission:
► Diagnosed with primary unilateral inguinal hernia
► Aged 18–80 years

- American Society of Anesthesiologists (ASA) classification I–II
- Provision of informed consent

## Exclusion criteria

Patients with any one or more of the following will be excluded from this study:
- Severe organ dysfunction or inability to tolerate surgery
- Hernia recurrence
- Giant hernia (inner size of the hernia >4 cm)
- Scrotal hernia
- Incarcerated inguinal hernia
- Inability to complete follow-up or questionnaire because of mental disorder or other reasons
- History of preperitoneal surgery, such as radical prostatectomy

## Randomisation and blinding

This study is a multicentre trial, so stratified block randomisation will be performed in each centre. A randomisation sequence table will be generated by a statician who will not be involved in the trial using Statistical Analysis System (SAS V.9.1). The serial numbers assigned to each patient will be preserved in opaque sealed envelopes. The sealed envelopes will be subsequently given to the trial centre. All patients will not know the surgical regimen until after the surgery. The surgeons will not be blinded to the surgical regimen. Outcome assessors will be blinded to the surgical records in the electronic case report form (eCRF).

## Interventions

Based on recommendations for treatment and follow-up of inguinal hernias in adults made by the Chinese Medical Association and Chinese Medical Doctor Association[17] and the European Hernia Society.[18]

### Anterior transversalis fascia repair

An oblique skin incision will be made parallel to the inguinal groove from the inner to the outer inguinal ring. Subcutaneous tissue will be dissected until the external oblique aponeurosis is reached. Haemostasis will be achieved by electric coagulation. A4–6 cm long skin incision will be made on the external oblique aponeurosis, starting from the pubic tubercle, to fully expose the pubic tubercle and the inner ring.

The lower part of the external oblique aponeurosis will be separated from the spermatic cord. The upper part of the external oblique aponeurosis will then be separated from the internal oblique aponeurosis and dissociated to a point 3 cm above the inguinal canal wall. The overlying spermatic cord and cremaster muscle will be lifted up and disassociated about 2 cm above the pubic tubercle. The ilioinguinal nerve, external spermatic vessels and genital nerves concomitant with the spermatic cord will be protected while the spermatic cord is lifted.

The inguinal canal will then be dissected and the hernia sac will be dissociated from the surrounding tissue. The cremaster muscle fibres will be separated longitudinally and the hernia sac will be separated from the spermatic cord. The hernia sac will be dissociated until the neck of the hernia sac is reached. All abdominal contents in the hernia sac will be returned to the abdominal cavity. In cases with a small hernia sac, the sac will also be returned to the anterior peritoneal cavity to minimise postoperative pain. In cases with a large hernia sac, the distal end will be transected and opened, and the proximal part will be ligated at a high position.

A patch designed with an arc-shaped head and a swallow tail-like end that comprises two pieces (a wide top piece and a narrow bottom piece) will be sutured and fixed with non-absorbable polypropylene sutures. The inner side of the patch will be sutured to the pubic tubercle and the lateral boundary of the rectus abdominis muscle. The upper border of the patch will be sutured to the conjoined tendon and its border with the inguinal ligament and iliopubic tract. The wound will then be closed using layered sutures according to anatomic position.

### Preperitoneal space repair

The skin will be cut and subcutaneous tissue will be dissected. The abdominal external oblique aponeurosis will be dissected along the fibres. The abdominal internal oblique aponeurosis and transversus abdominis muscle will be bluntly separated to expose the transverse fascia. Dissociation of the hernia sac will be done as described above in the anterior transversalis fascia repair.

Transabdominal preperitoneal or totally extraperitoneal laparoscopic techniques were used to dissociate preperitoneal space, strip spermatic cord peritoneum, anatomise and expose the pubic pore. A polypropylene patch will be placed in the prevesical space and Bogros' space to achieve repair of the pubic pore-containing area.

### Concomitant treatment

Any medications, with the exception of inguinal hernia-specific treatments, administered during hospitalisation will be recorded. Before starting the trial (ie, at the first visit), detailed information will be recorded regarding concomitant diseases, combined medication and measures to be taken. At discharge, changes in medications and measures to be taken will be recorded. For every combination of medication and measures to be taken, a minimum of the following information will be recorded: drug name (generic preferred), dosage, start date, stop date or continuing use and indications.

### Study flowchart
#### Before surgery
Visit 1 (at admission, day 0)
- Sign informed consent;
- Recheck inclusion/exclusion criteria;
- Demographic data (sex, age, body height, body mass index) and medical insurance type;

| Table 2 | Baseline information of patients with inguinal hernia |
|---|---|
| **Sex** | **Smoking history** |
| Age | History of alcohol use |
| Body height | Disease attack and admission |
| Body mass index | Laboratory examination |
| Medical insurance type | Imaging examination |
| Type of hernia | Vital sign |
| Inguinal hernia classification | VAS pain score |
| Treatment time | ECG |
| American Society of Anesthesiologist Classification | Concomitant therapy |
| History of diseases | |
| Diabetes mellitus | |
| Cardiovascular disease | |
| Lung disease | |
| Peripheral vascular disease | |
| Dementia | |
| Hypertension | |

VAS, Visual Analogue Scale.

▶ Type of hernia (indirect or direct);
▶ Inguinal hernia classification as type I, II, III or IV according to the adult inguinal hernia and femoral hernia classification (revised in 2003) developed by the Hernia and Abdominal Wall Surgery Group, Society of Surgery, Chinese Medical Association[19];
▶ ASA classification;
▶ History of diseases and risk factors* (diabetes mellitus, cardiovascular disease, lung disease, peripheral vascular disease, dementia, hypertension, smoking history and alcohol use history) (*optional evaluation items);
▶ Disease onset and admission (interval from first appearance of the lump, main symptoms);
▶ Imaging examination (ultrasound, CT);
▶ Laboratory examination (routine blood testing, coagulation testing, testing of blood glucose, lipids and electrolytes and hepatic and renal function);
▶ Vital signs (body temperature, pulse, respiration rate, blood pressure);
▶ VAS pain score;
▶ Electrocardiography;
▶ Concomitant treatment.

The baseline information of patients with inguinal hernia included in this study is shown in table 2.
Predischarge
Visit 2
▶ Vital signs (body temperature, pulse, respiration rate, blood pressure);
▶ Laboratory examination (routine blood testing, coagulation testing, testing of blood glucose, lipids and electrolytes and hepatic and renal function);

▶ Treatment regimen (inguinal hernia-specific treatment).

Surgical approach and patches (type and fixation method); anaesthesia method; drug name (generic preferred), dosage, route, start date, length of drug use will be recorded.
▶ Medical costs;
▶ Concomitant treatment.

Inguinal hernia-specific treatment, maintenance therapy and drugs used for concomitant treatment.
▶ Medical costs because of adverse events (AE).

### Follow-up
Visit 3 (1 week after surgery via clinic visit or telephone follow-up)
▶ 36-Item Short Form Health Survey (SF-36) score;
▶ Medical costs;
▶ Hernia recurrence (date and severity of recurrence). Recurrence will be diagnosed by the medical institutions, after being alerted by patients that symptoms are present.
▶ Presence or absence of haematoma and/or seroma;
▶ Wound infection (date and severity of wound infection).

Visit 4 (1 month after surgery via clinic visit or telephone follow-up)
▶ SF-36 score;
▶ Medical costs;
▶ Hernia recurrence (date and severity of recurrence). Recurrence will be diagnosed by the medical institutions, after being alerted by patients that symptoms are present.
▶ Presence or absence of haematoma and/or seroma;
▶ Wound infection (date and severity of wound infection).

Visit 5 (3 months after surgery via clinic visit or telephone follow-up)
▶ SF-36 score;
▶ Medical costs;
▶ Hernia recurrence (date and severity of recurrence). Recurrence will be diagnosed by the medical institutions, after being alerted by patients that symptoms are present.
▶ Presence or absence of haematoma and/or seroma;
▶ Wound infection (date and severity of wound infection).

Visit 6 (1 year after surgery via telephone follow-up)
▶ VAS pain score;
▶ SF-36 score;
▶ Medical costs;
▶ Hernia recurrence (date and severity of recurrence). Recurrence will be diagnosed by the medical institutions, after being alerted by patients that symptoms are present;
▶ Presence or absence of haematoma and/or seroma;

► Wound infection (date and severity of wound infection).

Visit 7 (2 years after surgery via telephone follow-up)
► VAS pain score;
► SF-36 score;
► Medical costs;
► Hernia recurrence (date and severity of recurrence). Recurrence will be diagnosed by the medical institutions, after being alerted by patients that symptoms are present;
► Presence or absence of haematoma and/or seroma;
► Wound infection (date and severity of wound infection).

## Outcome measures
### Primary outcome measure
To reduce the outcome bias of a single evaluation, the incidence of chronic pain at 1 and 2 years after surgery will be evaluated. According to International Association for the Study of Pain, VAS pain score >0 for over 3 successive months indicates chronic inguinal pain.[20]

### Secondary outcome measures
Postoperative complications including wound infection, rates of haematoma, seroma, hernia recurrence, intestinal obstruction, intestinal fistula, entocele, mesenteric thrombosis, femoral vein thrombosis, ischaemic orchitis, painful ejaculation, varicocele and scrotal oedema at postoperative 1 week, 1 and 3 months and 1 and 2 years.

### Other outcome measures.
Quality of life is evaluated by the SF-36.[21] The SF-36 is a 36-item, patient-reported survey of patient health. It consists of eight scaled scores, including vitality, physical functioning, bodily pain, general health perceptions, physical functioning, emotional functioning, social functioning and mental health. The score of each scale is summed and then standardised according to the formula: standardised score = (actual raw score—lowest possible raw score)/ possible raw score range×100. The total SF-36 score is the standardised score based on the sum of the eight scaled scores. A higher score indicates better quality of life.

The cost utility analysis of therapeutic regimens involving different surgical approaches will be analysed. Medical costs consist of direct medical and non-medical costs. The direct medical costs include drug charges, inspection fees, laboratory fees, treatment fees, nursing fees and bed charges. The direct medical costs during hospitalisation will be calculated according to the hospital information system. Direct medical costs during the follow-up period will be reported by patients and/or their relatives. Direct non-medical costs include payments for transportation to receive medical care, cost of nutritional supplementation and the costs for accompanying family members during the treatment period.

All outcome evaluations will be independently performed by an experienced assessor blinded to the treatment regimens. The schedule of outcome measurement assessment is shown in table 3.

### AE and serious AE
According to the study protocol and clinical judgement, AE/serious AE (SAE) occurring after herniorrhaphy will be reported to the Department of General Surgery, the Fourth Affiliated Hospital of China Medical University.

AE refer to any adverse medical events occurring after herniorrhaphy in patients or clinical subjects. SAE refer to any adverse postoperative medical events involving one or more of the following criteria:
► Death, irrespective of the cause;
► Life-threatening event;
► Severe or permanent disability or organ dysfunction;
► Haemorrhage;
► Malformation, birth defect;
► Hospitalisation or extended hospital stay;
► Readmission to hospital;
► Recurrence (symptomatic) of inguinal hernia.

### Causal relationship between surgery and adverse events
The causal relationship between the drugs used and AE will be evaluated by the researchers as: certainly relevant, probably relevant, likely relevant, unlikely relevant and irrelevant (table 4).

### Evaluation criteria for the severity of adverse events
Mild: The patient is aware of symptoms, but symptoms can be tolerated. Symptoms are causing mild discomfort but not interfering with daily activities.

Moderate: Not affecting daily activities.

Severe: Very painful, causing significant functional impairment or loss of self-care ability and prohibiting the patient from carrying out daily activities.

The researcher will evaluate the severity of AE according to clinical indices such as laboratory and inspection outcomes, not just based on the subject's direct feelings.

### AE reporting
Reporting time limit for AE will be within 24 hours of onset. In the clinical research period (from the time of signing informed consent to 1–2 years postoperatively), any AE occurring in any patient who received either surgical approach will be properly treated. According to the requirements of this study protocol and clinical evaluation, researchers will fill in the table of Adverse Events After Herniorrhaphy and submit this within 24 hours to the Clinical Research Center of Abdominal Wall Hernia, the Fourth Affiliated Hospital of China Medical University, China.

### Serious adverse event emergency reporting
Reporting time limit for SAE will be within 24 hours of onset. Any SAE will be continuously monitored and reported until it is healed, stabilises or recovers to near baseline conditions, irrespective of whether patients have terminated or completed treatment. Any follow-up information regarding SAE will be reported within 24 hours.

**Table 3** Timing of outcome measurement assessment

| | Before surgery | During surgery | Follow-up | | | | |
|---|---|---|---|---|---|---|---|
| | Visit 1 (at admission, day 0) | Visit 2 (at discharge) | Visit 3 (1 week after surgery) | Visit 4 (1 month after surgery) | Visit 5 (3 months after surgery) | Visit 6 (1 year after surgery) | Visit 7 (3 years after surgery) |
| Signed informed consent | X | | | | | | |
| Inclusion/exclusion criteria | X | | | | | | |
| Demographic data | X | | | | | | |
| Medical insurance type | X | | | | | | |
| Delayed visit and admission | X | | | | | | |
| Previous history of diseases | X | | | | | | |
| Previous history of drug | X | | | | | | |
| Risk factors | X | | | | | | |
| Disease attack and admission * | X | | | | | | |
| Type of hernia (indirect hernia, direct hernia) | X | | | | | | |
| Vital sign† | X | | | | | | |
| Laboratory examination ‡ | X | X | | | | | |
| Imaging examination § | X | | | | | | |
| Electrocardiography | X | | | | | | |
| Treatment regimen ¶ | | X | | | | | |
| VAS score ** | | | | | | X | X |
| SF-36 score | | | X | X | X | X | X |
| Medical cost | | X | X | X | X | X | X |
| Concomitant treatment | X | X | | | | | |
| Adverse events Wound infection Haematoma Seroma Hernia recurrence | | X | X | X | X | X | X |

*Indicates the interval from the first appearance of the lump or main symptoms.
†Indicates body temperature, pulse, respiration rate, and blood pressure.
‡Indicates routine blood testing, coagulation testing, testing of blood glucose, lipids and electrolytes and hepatic and renal function.
§Indicates ultrasound, CT examination.
¶Indicates inguinal hernia-specific treatment.
**Indicates the VAS pain score.
SF-36, 36-Item Short Form Health Survey; VAS, Visual Analogue Scale.

## Patient completion/withdrawal from clinical study

Patients for whom the whole clinical data of at least 1 year are collected will be considered as complete cases. Patients with any one or more of the following criteria will be considered withdrawn from the study: mistakenly recruited, withdrawal of informed consent, on the request of the sponsor for safety reasons or patient conflicts or lost to follow-up. The date and reasons for withdrawal will be recorded on the eCRF. After termination of the study, the data collected at the last visit will be evaluated, except for data from those lost to follow-up.

## Statistical analysis

Statistical analysis will be performed by a statician using SPSS V.19.0 software (SPSS). Continuous variables will be statistically described using the mean, SD, median, minimum and maximum. The categorical variables will be expressed using numbers and percentages.

The analysis will be performed on the basis of the intention-to-treat principle. Descriptive statistics of baseline information will be performed. The $\chi^2$ test or Fisher's exact test will be used for analysis of categorical variables, such as the incidence of postoperative chronic pain (primary outcome measure) and the incidence of postoperative

**Table 4** Causal relationship between surgery and adverse events

| | Certainly relevant | Probably relevant | Likely relevant | Unlikely relevant | Irrelevant |
|---|---|---|---|---|---|
| Adverse events are obviously caused by external factors | - | - | - | - | + |
| Adverse events are correlated with surgical treatment at rational time | + | + | + | - | - |
| Adverse events are correlated with patient diseases | - | - | + | + | + |
| Adverse events are correlated with suspected postoperative response patterns | + | +/- | + | - | - |
| After relief of related surgical factors, adverse events alleviate or disappear | + | + | - | - | - |
| After surgery-related factors worsen, adverse events recur | + | + | - | - | - |

To minimise the surgical risk and meet the requirements of laws and regulations, the sponsor will manage the correlations as follows: 'Irrelevant' belongs to the irrelevant category, and 'certainly relevant', 'probably relevant', 'likely relevant' and 'unlikely relevant' belong to the relevant category.

complications (secondary outcome measure) between groups. Independent sample t-test or Mann-Whitney U test will be used for comparisons of continuous variables, such as SF-36 score, between groups. Cost-utility analysis will be used for economic evaluation, and sensitivity analysis of cost and utility will be performed.

### Interim analysis

When an adequate number of patients are enrolled and followed-up, interim analysis will be performed. When data are included for the full analysis set and recorded in the database, the first interim analysis during the management period will be performed to check whether the core data collected are suitable for preliminary significant data analysis. According to research progression, subsequent interim analysis of all data included in the database will then be designated. After acquiring approval from the Department of General Surgery and Scientific Construction Committee, the Fourth Affiliated Hospital of China Medical University, the data collected in the Department of General Surgery, the Fourth Affiliated Hospital of China Medical University are likely to be analysed together with the data collected from the other research centres. In accordance with applicable laws and regulations, the information on the subjects in the study will be kept confidential. The data for interim analysis will be precisely described in a statistical analysis plan. The interim analysis results will be submitted to an independent data monitoring committee in the form of a statistical analysis report and as slides.

### Sample size

According to previous reports,[22 23] we hypothesised that the incidence of chronic inguinal pain after anterior transversalis fascia repair and preperitoneal space approach was 10% and 3.4%, respectively. Taking $\alpha=0.05$ and power=90%, the final effective sample size of n=600 was calculated. Assuming a patient loss rate of 20%, we require 720 patients.

### Ethics and dissemination

#### Ethical approval

Before study commencement, the following files will be provided to the Independent Ethics Committee (IEC):
► Final draft of study protocol (and supplements);
► Sponsor-approved informed consent and other documents provided to the subjects (such as participation card and diary card);
► Materials assisting patients to be included;
► Materials regarding study-related injury compensations or rewards for patient participation in the study;
► Researcher résumé or equivalent (unless the IEC states that this is not needed);
► Sponsor name, funds, potential competing interests and information that affects patient participation in the study;
► Any other documents required by the IEC.

Trials cannot be started until the IEC completely approves the study protocol, informed consent, materials assisting patients to be included and compensation measures for the patients, and the sponsor receives a copy of the IEC approval document (see online supplementary file 3). The IEC approval document should include the trial title (registration number), name of the study file (including edition code) and date of approval.

During the study period, it is likely that researchers will submit the following files to the IEC for approval at appropriate time-points:
► Supplement of study protocol;

- ► Informed consent forms and documents regarding rewards for patient participation in the study;
- ► New information that is likely to negatively affect participant safety and study progression;
- ► Files regarding bias and alterations of the study protocol made to avoid immediate injury to patients;
- ► Reports regarding dead patients;
- ► Notification of change of project manager;
- ► Other requirements of the IEC.

If the supplemented study protocol increases the risk to patients, the supplemented study protocol and corresponding modified informed consent form will be submitted to the IEC for consideration. The supplemented study protocol will not be performed until approval from the IEC is obtained. The major study protocol was approved by the IEC, the Fourth Affiliated Hospital of China Medical University (approval number 2015–027) on 27 November 2015. The study protocol should be reviewed by the IEC at least once every year, and the reviewed suggestion will be recorded on paper. At the end of the study, the researchers should inform the IEC of its completion.

We began recruitment in February 2017 and expect to have completed recruitment by December 2019 and completed data collection by June 2020. The final results of the trial will be published in international peer-reviewed journals.

### Informed consent to participate

Each patient (or his/her legal representative) will provide signed and dated informed consent before surgery after fully understanding the objective and contents of the study (see online supplementary file 4). The researcher or his/her authorised staff members will fully explain the objective, methods, possible benefits, potential risks and any possible discomforts of the study to the potential patients before inclusion. Participants will be informed that participation in the study is voluntary and that they can withdraw from the study at any time. The participants will know that their identifying information will be recorded for long-term follow-up and will be read by personnel from the related institutions and the sponsor within the permit of relevant laws and regulations. The right to privacy of the participant will be protected.

### Confidentiality

- ► Only data required to investigate the effectiveness and safety of herniorrhaphy will be collected and analysed.
- ► Data collection and use will not be disclosed to any non-authorised persons, and will be performed in accordance with the laws and regulations regarding protection of the participant's privacy.
- ► The process of data collection will be fair and lawful.
- ► The purpose of data collection will be specific, identified and legitimate, and the collected data will not be used for other unrelated objectives.
- ► The data collected will be adequate, related and not redundant relative to the study objective.

- ► The data collected will be accurate and updated when necessary.
- ► Before collection of personal data, researchers will obtain participant consent, which should include lay emphasis on the transfer of data to other institutional entities and countries.
- ► The participants have the right to obtain their data and can request to modify mistaken or incomplete data.
- ► During the study period, participant's personal information will not be obtained or disclosed to non-authorised persons and will not be illegally destroyed, lost or altered unexpectedly. During the entire study period, the sponsors who have the right to read the participant's personal information will keep the data confidential.

### Dissemination

Any unpublished information provided by the sponsors and all unpublished data relating to this study will be kept confidential and will be owned by the sponsor. This information or data will be not be used for other purposes unless written approval is acquired from the sponsor. The clinical researchers will be informed that study results will be used for further study. Therefore, the study results may be provided to other clinical researchers or related administrative departments. The study results will be disclosed in the form of a clinical study report, including data collected from any research centre involved. If clinical researchers publish the study outcomes, they will provide the original manuscript to the sponsor for online review 60 days before submission or presentation. A summary, posters or other promotional materials will be created to facilitate the review. The sponsor will discuss scientific and regulatory compliance issues with clinical researchers. The sponsor will not mandatorily require the clinical researchers to modify the scientific contents and has no right to hide information. The clinical researchers should consider the integrity of this multicentre study. The data from one research centre can be published only under the following circumstances: the articles involving the outcomes from all research centres have been published; study in all research centres has been accomplished, abandoned or terminated for 12 months; the sponsor has stated that they will not publish the study outcomes from multiple research centres. Assignment of the author listings in articles related to this study will be performed based on the author's contribution guidelines, such as the guidelines of Uniform Requirements for Manuscripts Submitted to medical journals.

### Data management
#### Protocol modification

Study protocol modification will be signed, dated and published by the Department of General Surgery, the Fourth Affiliated Hospital of China Medical University. The study protocol will not be put into clinical practice until IEC approval is received, unless this is necessary to

avoid risk to participants or to modify the study protocol regarding logistics and administration (for example, typographical errors and contradictions).

The study protocol should not be deviated from during clinical practice. When deviation exists, corresponding management will be performed. The causes and deviated contents will be recorded in the eCRF and original medical case notes. The study protocol deviation table and eCRF will be preserved in the research centre and sponsor institute.

### Participant identity registration and screening records

Participants must agree to fill in identification registration to enable individual identification of each participant. The monitor will recheck the integrity of this registration. The participant identity registration form will be confidential and will be preserved in the research centre. To ensure confidentiality, duplication of participant identity registration will be not permitted. All reports and letters relating to this study will be tagged with the relevant acronyms and serial number. The participant screening record form will be completed by doctors. The doctors will determine whether participants are eligible for admission to this study.

### Electronic case report form

In this study, the EDC will be used for data collection and management. All data relating to this study will be recorded on the eCRF provided by the sponsor. The researchers will fill in the eCRF after each participant visit, unless some clinical results cannot be acquired immediately. This ensures that the information recorded on the eCRF reflects the participant's latest outcome. Data accuracy will be performed by the researcher. Data recording, alteration and substitution will be performed by researchers or other authorised persons. All data inputs will be recorded to the EDC, and data queries will be made by researchers online via the EDC. The final data will not be altered and will be password protected.

### Data quality assurance

To ensure data accuracy and reliability, eligible researchers and appropriate research centres will be selected before study commencement. The monitor in the cosponsor research centre will monitor the study progression periodically. The cosponsor will advise the researchers how to fill in the eCRF. The monitor in the cosponsor research centre will visit the EDC to check the integrity and accuracy of the eCRF. Data recorded in the eCRF that is inconsistent with original data recorded will be altered by researchers or authorised persons.

### Auditing

Regular on-site inspection visits will be made by the cosponsors. The cosponsors' monitor will date the inspection on an inspection form, which will then be preserved in the research centre. After study commencement, the first on-site inspection visit will be performed as soon as possible after participant recruitment. During on-site inspection, the monitor will check the consistency of data recorded in the eCRF with original data recorded in medical notes from the research centre. The nature and preservation place of original data documents will be confirmed to enable clinical researchers to know the source of all original data required in the eCRF, thus the monitor of the cosponsor can recheck these data.

If original data are electronically preserved, the monitor of the cosponsor will discuss the recheck method with clinical researchers. The original data document will include participant identity, eligibility for inclusion, informed consent, dates of visits, execution of study protocol, curative effects, safety index, AE reporting and follow-up, medication and date of study completion.

The monitor of the cosponsor will discuss the detailed requirements for original data recording with clinical researchers. To recheck whether the data recorded in eCRF is consistent with original data, the monitor of the cosponsor will be provided with the required original data. The monitor will discuss any problems found during rechecking of data consistency with clinical researchers. The clinical researchers will regularly discuss the information feedback.

### Study completion/termination
#### Study completion

When the last visit of the last participant is completed, the research centre will inform the sponsor, and study completion will be designated. The sponsor will inform all research centres of the time of study completion. Further research after this time must be approved by the sponsor and can then be performed without protocol supplements.

#### Study termination

The sponsor will have the right to terminate the study at any research centre at any time possibly because of, but not limited to, the following criteria:
- ▶ The number of patients recruited reaches the predetermined requirements.
- ▶ Research cannot abide by the study protocol or GCP guidelines.
- ▶ Insufficient numbers of participants are recruited.

### Audit and inspection

A representative of the department of clinical quality assurance of the cosponsor may visit any of the research centres to determine whether the study protocol follows the laws and regulations. All study records, including original medical notes, will be disclosed to the representative. However, the privacy of the subject will be respected. The research centre will be informed about this visit in advance to allow sufficient time for appropriate preparation.

## DISCUSSION
### Significance of this study

This study will be the first large sample (720 patients), multicentre, randomised, controlled, clinical trial to

investigate the effectiveness of inguinal hernia repair via the anterior transversalis fascia approach versus the preperitoneal approach in patients from northern China. In this study, postoperative complications will be used as the primary outcome measure, and patient quality of life and cost-utility analysis will be the secondary outcome measures.

However, this study is also limited: (1) Although all surgeons involved in the study will receive training of standard surgical procedure, confounding variables including surgeon's clinical experiences and surgical conditions should be considered to reduce the bias in the estimate of the study outcomes. (2) The time taken for studying postoperative recurrence of inguinal hernia is limited, which likely influences the judgement of long-term recurrence of inguinal hernia.

## Contribution to future studies

Based on study results, we aim to effectively reduce physical and psychological pain, ensure high-quality medical care (including safety) and achieve the best rehabilitation in the treatment of inguinal hernia. We aim to determine how to reduce medical resources (including shortening treatment time and reducing labour service strength) and medical costs and improve the efficiency of medical work and other issues. This study will provide important clinical guidance as to the method of inguinal hernia repair that is most suitable for the anatomic characteristics of patients in northern China and adaptive to the regional economic situation.

## Trial status

Recruitment of patients is ongoing at the time of submission.

**Author affiliations**
[1]Department of General Surgery, The Fourth Affiliated Hospital of China Medical University, Shenyang, China
[2]Department of General Surgery, Shengjing Hospital of China Medical University, Shenyang, China
[3]Department of General Surgery, Fengtian Hospital of Shenyang Medical College, Shenyang, China
[4]Department of General Surgery, The First Hospital of Dandong City, Dandong, China
[5]Department of General Surgery, The First Affiliated Hospital of Dalian Medical University, Dalian, China
[6]Department of General Surgery, General Hospital of Benxi Steel and Iron (Group), Fifth Clinical College of China Medical University, Benxi, China
[7]Department of General Surgery, General Hospital of Shenyang Military Area, Shenyang, China
[8]Department of General Surgery, The 202nd Hospital of PLA, Shenyang, China
[9]Department of General Surgery, The First Affiliated Hospital of Liaoning Medical University, Jinzhou, China

**Contributors** HYL, FQY, SJZ, YQW, WDA, ZSD, HYJ, FRW and SFQ conceived the study and participated in its design and coordination. LY drafted the manuscript. SBW and QF participated in the design of the study and performed the statistical analysis. DWZ and DYY participated in the study design and coordination and helped draft the manuscript. HWL participated in the design of the study and wrote the protocol for the analysis. All authors read, revised and approved the final manuscript.

**Funding** This research was funded by National Natural Science Foundation N0.81472302.

**Competing interests** None declared.

**Patient consent** obtained.

**Ethics approval** The Ethics Committee of the Fourth Affiliated Hospital of China Medical University.

**Provenance and peer review** Not commissioned; externally peer reviewed.

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
