## [Reviewer comments · BMJ Open]

ARTICLE DETAILS

TITLE (PROVISIONAL)	Anterior transversalis fascia approach versus preperitoneal space approach for inguinal hernia repair in residents in Northern China: study protocol for a prospective, multi-center, randomized, controlled trial
AUTHORS	Fan, Qing; Zhang, Dewei; Yang, Daye; Li, Hongwu; Wei, Shibo; Yang, Liang; Yang, Fuquan; Zhang, Shaojun; Wu, Yaoqiang; An, Weide; Dai, Zhongshu; Jiang, Huiyong; Wang, Furong; Qiao, Shifeng; Li, Hangyu

VERSION 1 - REVIEW

REVIEWER	Professor Aali J Sheen Central Manchester NHS Foundation Trust, United Kingdom
REVIEW RETURNED	07-Apr-2017

GENERAL COMMENTS	1) needs a table of other randomised studies undertaken in this field2) References need updating as there are many RCTs in this area and they need to be cited e.g Shouldice v Lichtenstein, lap v open3) t test is only used for parametric data analysis, but statistics should be evaluated by a statistician4) Why do they think this RCT is necessary, comparing two open techniques5) They need to specify why they are not offering the patients laparoscopic surgery - this maybe a resource issue6) Will antibiotics be given routinely
---

REVIEWER	Maciej Pawlak Medical University of Gdańsk, Poland
REVIEW RETURNED	20-Apr-2017

GENERAL COMMENTS	This is a well design study protocol but there are no real data presented. Further there is a lack of up-to-date cited meta analysis published on the subject, for example: 1. Zhu X, Cao H, Ma Y, Yuan A, Wu X, Miao Y, et al. Totally extraperitoneal laparoscopic hernioplasty versus open extraperitoneal approach for inguinal hernia repair: a meta-analysis of outcomes of our current knowledge. Surg J R Coll Surg Edinburgh Irel. 2. Pisanu A, Podda M, Saba A, Porceddu G, Uccheddu A. Meta-analysis and review of prospective randomized trials comparing laparoscopic and Lichtenstein techniques in recurrent inguinal hernia repair. Hernia Nevertheless, I find the research more than interesting and essential for the subject and future analysis if the problem. I'm sure that the results will deserve publication in a prestigious surgical journal.
---

REVIEWER	Neil Scott University of Aberdeen, UK
REVIEW RETURNED	01-May-2017

GENERAL COMMENTS	This is a well-written protocol for a potentially important randomised trial of approaches to hernia repair. I have some comments about the methodology, although some of these may require changes to the original study protocol and not just this article. Both the method used to generate the random sequence and the method to conceal the next random allocation are not made clear. These are particularly important measures of quality in multicentre surgical trials as bias can be caused if the operator knows what the next allocation will be. It is implied that each centre will be performing randomisation locally instead of using a central randomisation service. A number of statistical tests are described, some to look at change from baseline and some to look at final scores, but some tests may be missing as I did not see a mention of a test for continuous data when examining final scores. The strategy to compare either change scores or final scores seems a bit confused - personally I would analyse final scores but with adjustment for baseline. The use of Bonferroni adjustment in the analysis is also rather unclear. There is mention of the main outcome (VAS pain) being analysed as both a continuous and binary outcome. Will the interim analysis be seen by an independent Data Monitoring Committee? This would not normally be seen by the project manager. It is not justified why 2000 patients will be recruited when the sample size necessary for the primary outcome is only 360. Apart from these points, I think this is a strong protocol that is suitable for publication.
--

VERSION 1 – AUTHOR RESPONSE

Reviewer: 1
 Professor Aali J Sheen
 Central Manchester NHS Foundation Trust, United Kingdom
 Please state any competing interests or state 'None declared': None declared

Please leave your comments for the authors below

1) needs a table of other randomised studies undertaken in this field
 Reply: We summarized randomized trials undertaken in this field in Table 1.

2) References need updating as there are many RCTs in this area and they need to be cited e.g
 Shouldice v Lichtenstein, lap v open
 Reply: We updated the references in Table 1.

3) t test is only used for parametric data analysis, but statistics should be evaluated by a statistician

Reply: We have revised the Statistical analysis section as follows:

Statistical analysis will be performed by a statistician using SPSS 19.0 software (SPSS, Chicago, IL, USA). Continuous variables will be statistically described using the mean, standard deviation, median, minimum and maximum. The categorical variables will be expressed using numbers and percentages. The analysis will be performed on the basis of the intention-to-treat principle. Descriptive statistics of baseline information will be performed. The chi-squared test or Fisher's exact test will be used for analysis of categorical variables, such as the incidence of postoperative chronic pain and the percentage of patients having postoperative complications between groups. Independent sample t-test or Mann-Whitney U test will be used for comparisons of continuous variables, such as SF-36 score, between groups. The cost-utility analysis will be performed for economic evaluation, and sensitivity analysis of cost and utility indicators will be also performed.

4) Why do they think this RCT is necessary, comparing two open techniques

Reply: The novelty of this study is to investigate the common postoperative complications of anterior transversalis fascia approach versus preperitoneal space approach for inguinal hernia repair in residents in Northern China. Chronic pain will be the largest area of interest, because we will mainly determine the optimal surgical hernia repair approach that is suitable for the anatomic characteristics of the inguinal region of residents in northern China and corresponds to the regional economic conditions. RCT is to reduce the effects of selection bias and measurement bias on result reliability.

5) They need to specify why they are not offering the patients laparoscopic surgery - this maybe a resource issue

Reply: Some patients will not be subjected to laparoscopic surgery because conventional celiotomy is mainly used in some centers involved in this study.

6) Will antibiotics be given routinely

Reply: No.

Reviewer: 2

Maciej Pawlak

Medical University of Gdańsk, Poland

Please state any competing interests or state 'None declared': None declared

Please leave your comments for the authors below

This is a well design study protocol but there are no real data presented. Further there is a lack of up-to-date cited meta analysis published on the subject, for example:

1. Zhu X, Cao H, Ma Y, Yuan A, Wu X, Miao Y, et al. Totally extraperitoneal laparoscopic hernioplasty versus open extraperitoneal approach for inguinal hernia repair: a meta-analysis of outcomes of our current knowledge. Surg J R Coll Surg Edinburgh Irel.

2. Pisanu A, Podda M, Saba A, Porceddu G, Uccheddu A. Meta-analysis and review of prospective randomized trials comparing laparoscopic and Lichtenstein techniques in recurrent inguinal hernia repair. Hernia

Reply: We have revised the body text and cited two lines of references [9][10].

Nevertheless, I find the research more than interesting and essential for the subject and future analysis if the problem. I'm sure that the results will deserve publication in a prestigious surgical journal.

Reply: thanks for your review.

Reviewer: 3

Neil Scott

University of Aberdeen, UK

Please state any competing interests or state 'None declared': None declared

Please leave your comments for the authors below

This is a well-written protocol for a potentially important randomised trial of approaches to hernia repair.

I have some comments about the methodology, although some of these may require changes to the original study protocol and not just this article.

Both the method used to generate the random sequence and the method to conceal the next random allocation are not made clear. These are particularly important measures of quality in multicentre surgical trials as bias can be caused if the operator knows what the next allocation will be. It is implied that each centre will be performing randomisation locally instead of using a central randomisation service.

Reply: In the randomization and blinding section, we revised the content as follows:

This study is a multi-center trial, so stratified block randomization will be performed in each center. A randomization sequence table will be generated by a statistician who will not be involved in the trial using Statistical Analysis System (SAS 9.1). The serial numbers assigned to each patient will be preserved in opaque sealed envelopes. The sealed envelopes will be subsequently given to the trial center. All patients will not know the surgical regimen until after the surgery. The surgeons will not be blinded to the surgical regimen. Outcome assessors will be blinded to the surgical records in the electronic case report form (eCRF).

A number of statistical tests are described, some to look at change from baseline and some to look at final scores, but some tests may be missing as I did not see a mention of a test for continuous data when examining final scores. The strategy to compare either change scores or final scores seems a bit confused - personally I would analyse final scores but with adjustment for baseline. The use of Bonferroni adjustment in the analysis is also rather unclear.

There is mention of the main outcome (VAS pain) being analysed as both a continuous and binary outcome.

Reply: In this study, we will take the incidence of chronic pain 1 and 2 years after surgery as the primary outcome measure and we have revised the corresponding contents in the Statistical analysis section

Will the interim analysis be seen by an independent Data Monitoring Committee? This would not normally be seen by the project manager.

Reply: In the Interim analysis section, we revised it as follows: "The interim analysis results will be submitted to an independent Data Monitoring Committee in the forms of Second Assessment Report (SAR) and slideshow."

It is not justified why 2000 patients will be recruited when the sample size necessary for the primary outcome is only 360.

Reply: The value of the power was up-regulated and the sample size $n = 720$ was obtained. Sample size $n = 720$ will be used in our study.